# A Transcription Factor Regulates Gene Expression in Chloroplasts

**DOI:** 10.3390/ijms22136769

**Published:** 2021-06-24

**Authors:** Kexing Xin, Ting Pan, Shan Gao, Shunping Yan

**Affiliations:** College of Life Science and Technology, Huazhong Agricultural University, Wuhan 430070, China; xinkexing917@webmail.hzau.edu.cn (K.X.); panting@mail.hzau.edu.cn (T.P.); 2019gaoshan@webmail.hzau.edu.cn (S.G.)

**Keywords:** chloroplast, gene expression, NAC102, nucleus, transcription factor

## Abstract

The chloroplast is a semi-autonomous organelle with its own genome. The expression of chloroplast genes depends on both chloroplasts and the nucleus. Although many nucleus-encoded proteins have been shown to localize in chloroplasts and are essential for chloroplast gene expression, it is not clear whether transcription factors can regulate gene expression in chloroplasts. Here we report that the transcription factor NAC102 localizes in both chloroplasts and nucleus in *Arabidopsis*. Specifically, NAC102 localizes in chloroplast nucleoids. Yeast two-hybrid assay and co-immunoprecipitation assay suggested that NAC102 interacts with chloroplast RNA polymerases. Furthermore, overexpression of NAC102 in chloroplasts leads to reduced chloroplast gene expression and chlorophyll content, indicating that NAC102 functions as a repressor in chloroplasts. Our study not only revealed that transcription factors are new regulators of chloroplast gene expression, but also discovered that transcription factors can function in chloroplasts in addition to the canonical organelle nucleus.

## 1. Introduction

One of the major differences between plants and animals is that plants contain chloroplasts, which carry out photosynthesis, produce various metabolites, and sense external cues. Chloroplasts are believed to derive from cyanobacterium in an endosymbiotic event. During evolution, most bacteria genes were transferred to the host nucleus and only a small portion of genes were retained on the own genome [1,2]. Therefore, the crosstalk between chloroplasts and the nucleus is essential to maintain cellular homeostasis, which is achieved through anterograde (nucleus-to-chloroplasts) and retrograde (chloroplasts-to-nucleus) signaling [3,4].

As a semi-autonomous organelle, the gene expression in chloroplasts is controlled by both chloroplasts and the nucleus [2]. For example, both plastid-encoded RNA polymerase (PEP) and the nuclear-encoded RNA polymerase (NEP) contribute to chloroplast gene expression [5,6]. While the expression of photosynthetic genes, such as *psbA*, *psbB,* and *rbcL*, is preferentially dependent on PEP, the expression of the housekeeping genes, including ribosomal RNAs and the core subunits of the PEP, is dependent on NEP [7,8,9]. NEP is a phage-type RNA polymerase with a single subunit, encoded by two nuclear genes, *rpoTp* and *rpoTmp,* in *Arabidopsis*. PEP is a bacteria-type RNA polymerase that consists of four core subunits (α, β, β′, and β″), encoded by *rpoA*, *rpoB*, *rpoC1*, and *rpoC2* in the chloroplast genome, and a promoter-recognizing sigma factor (*σ*) encoded by the nuclear genes [10,11]. Furthermore, PEP forms a big complex with many PEP-associated proteins (PAPs) to regulate gene expression. All PAPs are encoded by the nuclear genome [12,13,14].

In addition to PAPs, many other nucleus-encoded proteins have been reported to regulate chloroplast gene expression at the transcriptional level or the post-transcriptional level (including RNA splicing, RNA editing, and translation) [13,15,16]. Although much progress has been made towards understanding their functions in post-transcriptional regulation, how they regulate chloroplast gene expression at the transcriptional level is less well-understood.

In plants, the NAC (NAM, ATAF, and CUC) family is one of the largest plant-specific transcription factors and plays important roles in plant development and stress responses [17,18]. Previous studies have revealed that the transcription factor NAC102 is involved in oxidative stress responses and the *nac102* mutant is sensitive to excessive light [19,20,21]. Here we show that NAC102 associates with the chloroplast genome, interacts with chloroplast RNA polymerases, and regulates chloroplast gene expression. To our knowledge, NAC102 is the first transcription factor reported to regulate chloroplast gene expression directly. Our study not only revealed that transcription factors are new regulators of chloroplast gene expression, but we also discovered that transcription factors can function in chloroplasts in addition to the canonical organelle nucleus.

## 2. Materials and Methods

### 2.1. Plant Materials and Growth Condition

All *Arabidopsis thaliana* used in this study were in the Columbia (Col-0) background. The *nac102* mutant (SALK_030702) was obtained from ABRC. The transgenic lines were generated by the floral dip method [22]. *Arabidopsis* was grown on half-strength (1/2) Murashige and Skoog (MS) medium or in soil under long-day conditions (16 h of light and 8 h of dark) at 22 ± 2 °C.

### 2.2. Vector Constructions

For localization analysis of NAC102, the coding sequence of enhanced yellow fluorescent protein (eYFP) was fused to the C-terminus of the NAC102 fragment (NAC102-eYFP) using fusion PCR. The NAC102^ΔN43^-eYFP was generated by using NAC102-eYFP as a template. To generate NAC102^N43^-eYFP, NAC102^N60^-eYFP, and NAC102^N80^-eYFP, the 43, 60 and 80 aa N-terminal parts of NAC102 were cloned and fused to eYFP, respectively. To generate NAC102-eYFP^NES^, the coding sequence of nuclear export sequence (NES) (CLPPLERLTLD) [23] was further fused C-terminally to NAC102-eYFP (Figure 1). To generate TRXz-CFP, the coding sequence of cyan fluorescent protein (CFP) was fused to the C-terminus of *TRXz*. All fusion fragments were inserted into pFGC5941 at the *Nco*I and *Xba*I sites. For Y2H assays, the coding sequence of NAC102 was inserted into pGBKT7 at the *Nde*I and *Pst*I sites. The coding sequences of the subunits of PEP and NEP were inserted into pGADT7 at the *Nde*I and *Pst*I sites. All recombination reactions were performed using a Lightening Cloning kit (Biodragon, Beijing, China). The primers used for vector constructions are listed in the Appendix A, Appendix A.

### 2.3. Transient Expression

For transient expression in *Arabidopsis* protoplasts, the plasmids were transfected into protoplasts as described previously [24] and incubated under dark for 16–24 h before imaging.

### 2.4. Confocal Laser Scanning Microscopy

Images were captured using a confocal laser scanning microscope (Leica TCS SP8 or DMi8, Wetzlar, Germany). The excitation wavelengths for DAPI, CFP, eYFP, and chlorophyll autofluorescence were 405 nm, 448 nm, 488 nm, and 514 nm, respectively. The emission filters were substrate for DAPI, 448–514 nm for CFP and eYFP, and 500–530 nm for chlorophyll autofluorescence.

### 2.5. Y2H Assay

The pGADT7 vectors were transformed into yeast strain AH109, while the pGBKT7 vectors were transformed into yeast strain Y187. After mating AH109 with Y187, the yeasts were plated on DDO media. The colonies were resuspended and diluted using distilled water, and were then plated on DDO or QDO media, respectively. The photos were taken after incubation at 28 °C for 2 d on DDO media or 3 d on QDO media.

### 2.6. CoIP Assay

The 8-day-old seedlings were ground in liquid nitrogen and incubated with IP buffer (50 mM Tris-HCl pH 7.4, 150 mM NaCl, 1% Triton X-100, 1% sodium deoxycholate, 20 mM EDTA, 1 mM PMSF, 100 µM MG132, and 1× protease cocktail) for 30 min. After centrifugation at 12,000× *g* for 5 min, the resulting supernatants were incubated with 40 µL GFP-Trap beads (Chromotek, Planegg-Martinsried, Germany) for 1 h. After washing three times with washing buffer (50 mM Tris-HCl pH 7.4, 150 mM NaCl, 1% Triton X-100, 1% sodium deoxycholate, 0.1% SDS, and 20 mM EDTA), the beads were incubated with 40 µL 2× SDS-PAGE buffer for 10 min at 98 °C. The eluted proteins were subjected to Western blotting using anti-GFP (Roche, Basel, Switzerland, 11814460001), anti-rpoA (PhytoAB, San Jose, USA, PHY1241S), anti-rpoB (PhytoAB, PHY1701), anti-rpoC1 (PhytoAB, PHY1240), anti-rpoC2 (PhytoAB, PHY0382A), anti-rpoTp (PhytoAB, PHY0836S), and anti-rpoTmp (PhytoAB, PHY0838S).

### 2.7. RNA Isolation and qPCR

Total RNA was extracted from 8-day-old *Arabidopsis* seedlings using TRIZOL reagent (Thermo Fisher Scientific, Carlsbad, USA). The reverse-transcription reaction was performed using HiScript II Q RT SuperMix (Vazyme, Nanjing, China) according to the manufacturer’s instructions. The qPCR assays were performed on the CFX96 Touch Real-Time PCR Detection System (Bio-Rad, Hercules, CA, USA) using ChamQ Universal SYBR qPCR Master Mix (Vazyme, Nanjing, China). The primers used for qPCR analyses are listed in Appendix A.

### 2.8. Chlorophyll Content Measurement

Chlorophyll contents were measured as described previously [25]. The 3rd and 4th leaves of 3-week-old seedlings were used.

## 3. Results

### 3.1. NAC102 Localizes in Both Chloroplasts and Nucleus

In *Arabidopsis thaliana*, there are 105 NAC transcription factors [26], among which NAC102 (*AT5G63790*) is predicted to localize in both chloroplast and nucleus according to TAIR (https://www.arabidopsis.org; 1 June 2016). To our knowledge, no transcription factors have been reported to localize in the chloroplast. Therefore, we decided to study NAC102 further. *NAC102* has two alternative transcripts, *AT5G63790.1* and *AT5G63790.2.* As shown in Appendix A, compared to *AT5G63790.1*, *AT5G63790.2* encodes a protein with an additional 10 amino acids at the N-terminus. To confirm the subcellular localization of NAC102, both NAC102 variants were fused with enhanced yellow fluorescent protein (eYFP) and were transiently expressed in *Arabidopsis* protoplasts. The CFP-tagged E2Fa transcription factor (E2Fa-CFP) was used as the nucleus marker and the chlorophyll fluorescence was used as the chloroplast marker. As shown in Figure 2, both AT5G63790.1-eYFP and AT5G63790.2-eYFP localized in chloroplast and nucleus. In the following study, we only used *AT5G63790.1.* To further confirm its localization, we generated the transgenic *Arabidopsis* expressing *NAC102-eYFP* (*AT5G63790.1*) driven by the 35S promoter (*35S:NAC102-eYFP*). Consistently, the eYFP fluorescence was detected in both chloroplast and nucleus (Figure 2C).

### 3.2. The N-Terminal Sequence of NAC102 Is Necessary and Sufficient for Its Chloroplast Localization

Most nucleus-encoded chloroplast proteins contain transit peptides at their N-terminus to direct them to the chloroplast. To map the transit peptide of NAC102, we performed a prediction using the ChloroP server [27]. However, no transit peptide was identified. Interestingly, when the NAC102 orthologs from other plant species were aligned, we found that the N-terminal 43 amino acid residues (N43) of NAC102 from *Arabidopsis thaliana* is very unique (Appendix A). We speculated that N43 is necessary for its chloroplast localization. To test this hypothesis, we transiently expressed the truncated NAC102-eYFP without N43 (NAC102^ΔN43^-eYFP) in *Arabidopsis* protoplasts. In support of the hypothesis, the NAC102^ΔN43^-eYFP was only detected in the nucleus (Figure 3A). To investigate whether N43 is sufficient for its chloroplast localization, we fused eYFP with N43 (N43-eYFP) and examined its localization. As shown in Figure 3B, the N43-eYFP could not localize in chloroplasts, suggesting that more amino acid residues are required for the chloroplast localization of NAC102. Therefore, we investigated the localization of eYFP fused to N-terminal 60 or 80 amino acid residues (N60 or N80). We found that the N80-eYFP but not N60-eYFP could localize in chloroplasts (Figure 3C,D). These results suggested that the N-terminal sequence of NAC102 is necessary and sufficient for its chloroplast localization.

### 3.3. NAC102 Is in Chloroplast Nucleoids

In chloroplast, NAC102-eYFP forms puncta (Figure 2), which is reminiscent of chloroplast nucleoids [28]. To test whether NAC102 is in chloroplast nucleoids, we isolated the chloroplasts from *NAC102-eYFP* transgenic lines and stained them with DNA-specific dye, DAPI. As shown in Figure 4A, the eYFP and DAPI fluorescence largely overlapped. TRXz (AT3G06730) is a well-characterized protein localized in chloroplast nucleoids [29,30]. Therefore, we fused TRXz with CFP to generate TRXz-CFP and co-expressed it with NAC102-eYFP in *Arabidopsis* protoplasts. We found that the fluorescence of NAC102-eYFP and TRXz–CFP co-localized in chloroplasts (Figure 4B). These results supported that NAC102 localizes to chloroplast nucleoids.

### 3.4. NAC102 Interacts with Chloroplast RNA Polymerases

Transcription factors regulate gene expression by interacting with RNA polymerases [31]. In chloroplasts, there are two types of RNA polymerases, PEP and NEP. To test whether NAC102 interacts with PEP and NEP, we carried yeast two-hybrid (Y2H) assays. The coding sequence of *NAC102* was fused to the GAL4 DNA-binding domain (BD) as a bait. The coding sequences of the subunits of PEP (rpoA, rpoB, rpoC1, and rpoC2) and NEP (rpoTp and rpoTmp) were fused to the GAL4 activation domain (AD) as prey. Compared with the AD control, the expression of PEP and NEP subunits promoted yeast growth on the selection media (QDO + 3-AT), indicating that NAC102 interacts with PEP and NEP subunits (Figure 5A). To confirm these interactions, we carried out co-immunoprecipitation (CoIP) assays using the NAC102-eYFP transgenic lines. In these assays, the transgenic line expressing the small subunit of ribulose-1,5-bisphosphate carboxylase/oxygenase and eYFP fusion (SSU-eYFP) was used as a negative control. Total proteins were extracted and subjected to immunoprecipitation with GFP-Trap. Both inputs and IP fractions were subjected to immunoblotting analysis with specific antibodies against rpoA, rpoB, rpoC1, rpoC2, rpoTp, and rpoTmp. Compared with SSU-eYFP, NAC102-eYFP could coimmunoprecipitate rpoA, rpoB, rpoTp, and rpoTmp, but not rpoC1 and rpoC2 (Figure 5B).

### 3.5. NAC102 Represses the Expression of Chloroplast Genes

Since NAC102 associates with the chloroplast genome and interacts with chloroplast RNA polymerases, we hypothesized that NAC102 may regulate the transcription of chloroplast genes. We first tested the chloroplast gene expression in the *nac102* mutant. However, the genes tested showed similar expression levels in Col-0 and *nac102* (Appendix A). Therefore, we next sought to test whether overexpression of NAC102 affects chloroplast gene expression. Given that NAC102 localized both in chloroplast and nucleus (Figure 2), it is possible that the nucleus fraction of NAC102 also regulates the transcription of chloroplast genes indirectly. To exclude the effect of nuclear NAC102, we generated transgenic lines over-expressing NES tagged NAC102-eYFP (NAC102-eYFP^NES^), which localized in chloroplasts, but not in the nucleus (Appendix A). As shown in Figure 5C,D, compared to Col-0 control, the expression of both the PEP (Figure 5C) and NEP-dependent (Figure 5D) genes was reduced in the *NAC102-eYFP^NES^* overexpression lines, suggesting that NAC102 represses the expression of these genes. In accordance with reduced chloroplast gene expression, we found that the chlorophyll contents were also reduced in the *NAC102-eYFP^NES^* overexpression lines (Figure 5E).

## 4. Discussion

Many studies have shown that the chloroplast gene expression depends on nucleus-encoded proteins. For example, the nucleus-encoded WHY1 and SIB1 have been shown to play important roles in chloroplast gene expression [32,33,34,35]. However, these proteins are transcriptional coregulators, but not transcription factors. In this study, we found that the transcription factor NAC102 associated with chloroplast genome and repressed chloroplast gene expression. To our knowledge, NAC102 is the first transcription factor reported to regulate chloroplast gene expression directly. Using ChloroP server [27], we found that many other transcription factors such as ARF2 (AT5G62000), YABBY1 (AT2G45190), WRKY3 (AT2G03340), bHLH014 (AT4G00870), and bZIP52 (AT1G06850) are predicted to localize in the chloroplast as well as in the nucleus. It is possible that these transcription factors also regulate chloroplast gene expression. Therefore, our study may encourage a new research direction to investigate the roles of transcription factors in chloroplast gene expression. Since transcription factors are well-known to regulate nuclear gene expression, our study indicated that transcription factors can function in chloroplasts in addition to the canonical organelle nucleus.

The crosstalk between chloroplast and nucleus plays an important role in plant growth and stress responses [3,4]. Since NAC102 localizes in both chloroplast and nucleus (Figure 2), it may function in both anterograde signaling and retrograde signaling. In the nucleus, NAC102 may regulate the expression of chloroplast-targeted genes. In chloroplast, NAC102 may regulate retrograde signals directly or indirectly. It is also possible that NAC102 itself functions as a retrograde signal, translocating from chloroplast to nucleus in certain conditions, which is worthwhile studying in the future.

Our data suggested that NAC102 represses chloroplast gene expression. When NAC102 was overexpressed in chloroplasts, both chloroplast gene expression and the chlorophyll contents were reduced (Figure 4). However, the chloroplast gene expression was not significantly different between Col-0 and *nac102* (Appendix A). One possible reason is functional redundancy. In addition to NAC102, other chloroplast-localized transcription factors may also regulate chloroplast gene expression. When NAC102 is mutated, other transcription factors may compensate for the role of NAC102. Previously, it was reported that the *nac102* mutant was more sensitive to excessive light stress [21]. Although we could not reproduce these results, it is possible that the chloroplast gene expression and the chlorophyll contents were higher in *nac102* in the experiment conditions, resulting in more light energy absorbed by chlorophyll and production of more reactive oxygen species (ROS), which triggered cell death. Although NAC102 functions as a repressor, other transcription factors in chloroplast may function as activators. Therefore, our study suggests that it is possible to regulate chloroplast gene expression using transcription factors, which may represent a new strategy to manipulate chloroplasts.

## Figures and Tables

**Figure 1 ijms-22-06769-f001:**
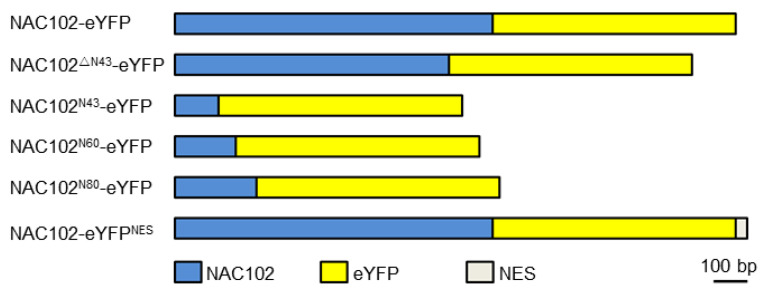
A schematic diagram of NAC102 fusion proteins.

**Figure 2 ijms-22-06769-f002:**
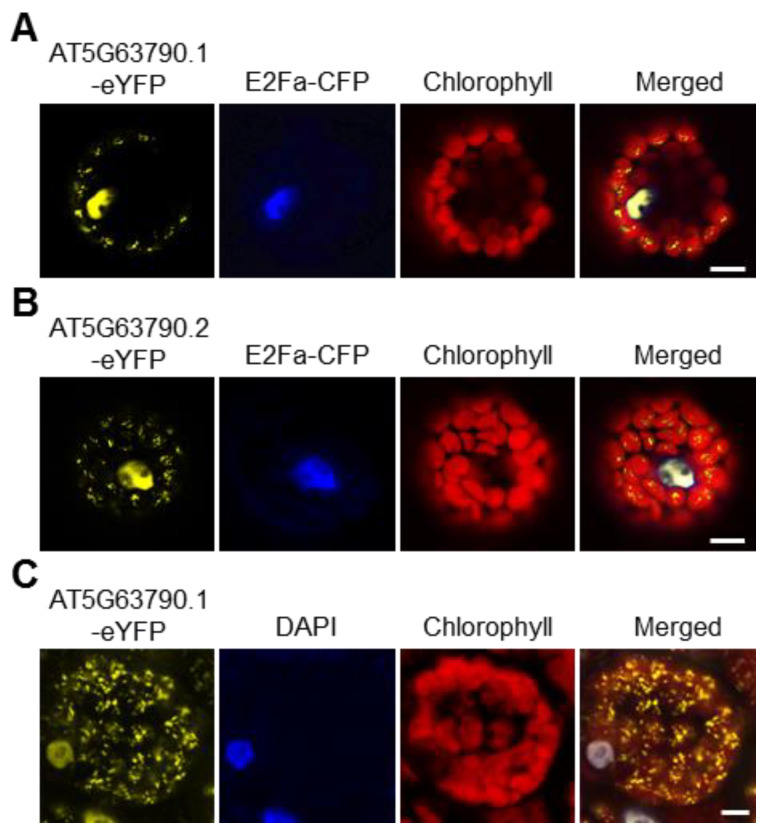
NAC102 localizes in both the nucleus and the chloroplast. (**A**,**B**) The localization of NAC102-eYFP when it was transiently expressed in *Arabidopsis* protoplasts. E2Fa-CFP was used as the nucleus marker and the chlorophyll fluorescence was used as the chloroplast marker. Both isoforms of NAC102, AT5G63790.1 (**A**) and AT5G63790.2 (**B**), were shown. (**C**) The localization of NAC102-eYFP in epidermal cells derived from NAC102 (AT5G63790.1) transgenic lines. The nucleus was stained with DAPI. The pictures were captured using confocal microscopy. Bars = 10 µm.

**Figure 3 ijms-22-06769-f003:**
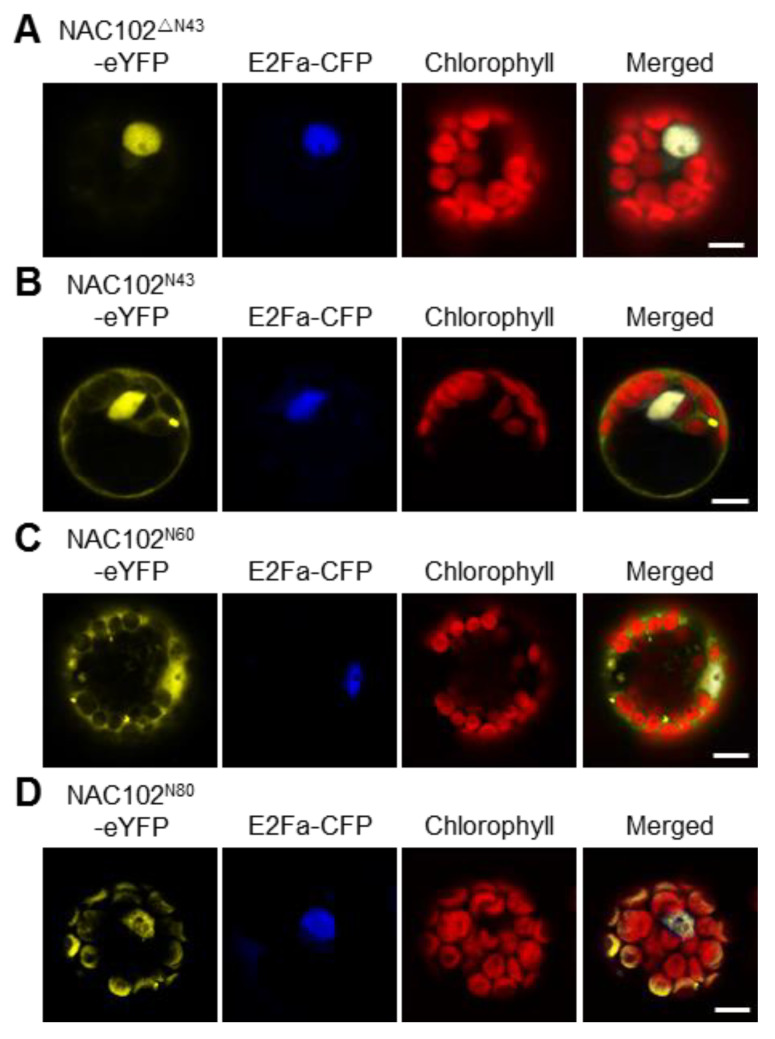
The effects of the N-terminal sequence of NAC102 on its chloroplast localization. The fusion proteins were transiently expressed in *Arabidopsis* protoplasts. NAC102^ΔN43^-eYFP (**A**) represented the truncated NAC102-eYFP without N-terminal 43 amino acid residues (N43); NAC102^N43^-eYFP (**B**), NAC102^N60^-eYFP (**C**), and NAC102^N80^-eYFP (**D**) represented eYFP fused with N-terminal 43, 60 or 80 amino acid residues, respectively. E2Fa-CFP was used as the nucleus marker and the chlorophyll fluorescence was used as the chloroplast marker. The pictures were captured using confocal microscopy. Bars = 10 µm.

**Figure 4 ijms-22-06769-f004:**
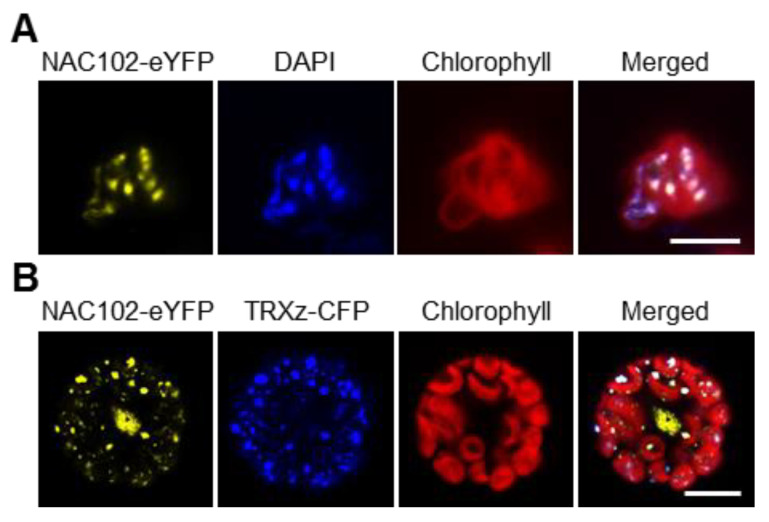
NAC102 associates with chloroplast nucleoids. (**A**) Colocalization of NAC102-eYFP and chloroplast nucleoids. The chloroplasts were isolated from *NAC102-eYFP* transgenic lines and stained with DAPI. The pictures were captured using confocal microscopy. Bar = 5 µm. (**B**) Co-localization of NAC102-eYFP and chloroplast nucleoid marker TRXz–CFP. The *NAC102-eYFP* and *TRXz-CFP* were transiently co-expressed in *Arabidopsis* protoplasts. The pictures were captured using confocal microscopy. Bar = 10 µm.

**Figure 5 ijms-22-06769-f005:**
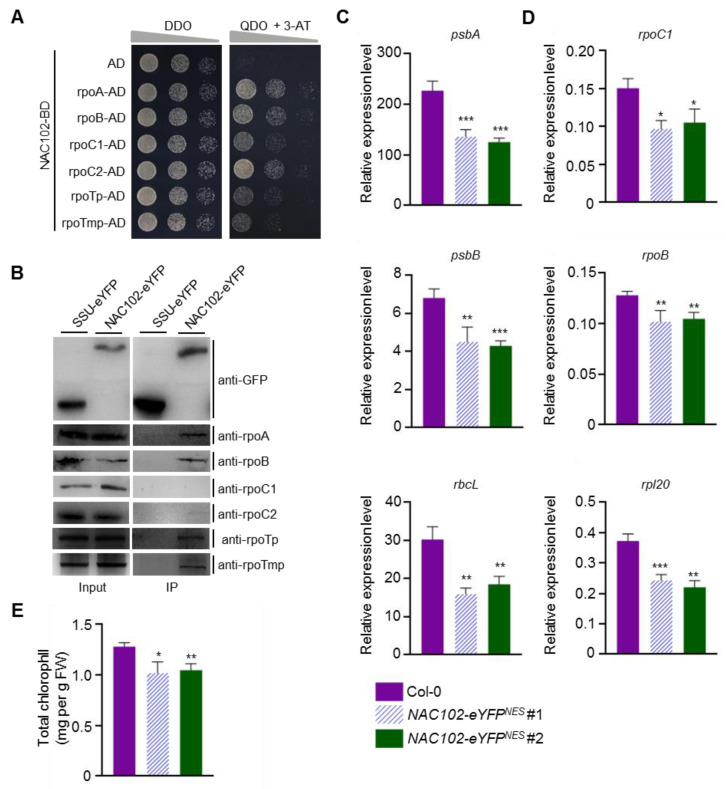
NAC102 interacts with chloroplast RNA polymerases to represses the expression of chloroplast genes. (**A**) Y2H assays. The serially diluted diploid cells were plated on DDO (lacking Trp and Leu) and QDO (lacking Trp, Leu, His, and Ade) supplemented with 10 mM 3-AT. The yeast growth on QDO + 3-AT indicates interaction. AD, activation domain. BD, DNA-binding domain. (**B**) CoIP assays. The total proteins of the NAC102-eYFP and SSU-eYFP transgenic lines were immunoprecipitated by GFP-Trap, followed by Western blotting using antibodies against rpoA, rpoB, rpoC1, rpoC2, rpoTp, and rpoTmp. The CoIP assays were repeated twice with similar results. (**C**,**D**) The relative expression level of the chloroplast genes determined by qRT-PCR analysis using *EF1α* as the normalizer. (**C**) The PEP-dependent genes. (**D**) The NEP-dependent genes. (**E**) Total chlorophyll content in the indicated plants. The 3rd and 4th leaves were used. Data represent mean ± SD (n = 3). The statistical significances were determined using Student’s *t*-test (* *p* < 0.05, ** *p* < 0.01, and *** *p* < 0.001).

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
