# Peer review of "A Transcription Factor Regulates Gene Expression in Chloroplasts"

_ijms, 2021, doi:10.3390/ijms22136769_

Round 1
Reviewer 1 Report
In the manuscript entitled with “A transcription factor regulates gene expression in chloroplasts” the authors performed good quality analysis regarding to the function of the NAC 102 transcription factor in chloroplasts. The authors demonstrated the localization of NAC 102 in both chloroplasts and the cell nucleus, as well as they analysed a potential sequence targeting chloroplasts (using transient expression in plant protoplast). The authors also shown that NAC102 localizes with chloroplast nucleoids.
I suggest that some parts of the “Results” should be removed to the discussion part.
The part” Vector constructions” should be better described, considering all gene constructs. Perhaps a schematic diagram could be used.
I need the explanation of the sentences “For transient expression assays, the coding sequence of eYFP was fused to the C terminal of different fragments of NAC102 using fusion PCR. The NAC102-eYFPNES was generated by PCR using NAC102-eYFP as the template and a reverse primer with the coding sequence of NES (CLPPLERLTLD)”.
Some sentences should be rearranged e.g. “To confirm the subcellular localization of NAC102, we fused both NAC102 variants with Enhanced Yellow Fluorescent Protein (eYFP) and were transiently expressed in Arabidopsis protoplasts.”
Author Response
Reviewer: 1
Comments and Suggestions for Authors
In the manuscript entitled with “A transcription factor regulates gene expression in chloroplasts” the authors performed good quality analysis regarding to the function of the NAC 102 transcription factor in chloroplasts. The authors demonstrated the localization of NAC 102 in both chloroplasts and the cell nucleus, as well as they analysed a potential sequence targeting chloroplasts (using transient expression in plant protoplast). The authors also shown that NAC102 localizes with chloroplast nucleoids.
Response: Thank you for your support.
I suggest that some parts of the “Results” should be removed to the discussion part.
Response: Thank you for your suggestion. In the “Results” part, we only explained the logic and described the data. We did not discuss the results in this part. Therefore, we didn't change the main content of the “Results” and “discussion” sections.
The part” Vector constructions” should be better described, considering all gene constructs. Perhaps a schematic diagram could be used.
Response: Thank you for your suggestion. We have revised this part with more details and all gene constructs were considered. We also drew a schematic diagram (Figure 1) to show the NAC102 fusion proteins.
I need the explanation of the sentences “For transient expression assays, the coding sequence of eYFP was fused to the C terminal of different fragments of NAC102 using fusion PCR. The NAC102-eYFPNES was generated by PCR using NAC102-eYFP as the template and a reverse primer with the coding sequence of NES (CLPPLERLTLD)”.
Response: Thank you for pointing it out. To generate NAC102-eYFP, the first fragment was amplified using NAC102 as the template with the forward primer targeting 5’ end of NAC102 and the reverse primer targeting 3’ end of NAC102 and 5’ end of eYFP. The second fragment was amplified using the eYFP as template with forward primer targeting 3’ end of NAC102 and 5’ end of eYFP and reverse primer targeting 3’ end of eYFP. NAC102-eYFP was amplified using the pool of the first fragment and the second fragment as the template using the forward primer targeting 5’ end of NAC102 and the reverse primer targeting 3’ end of eYFP.
NAC102-eYFPNESwas amplified using NAC102-eYFP as the template with the forward primer targeting the 5’ end of NAC102 and the reverse primer targeting 3’ end of eYFP and NES sequence.
Some sentences should be rearranged e.g. “To confirm the subcellular localization of NAC102, we fused both NAC102 variants with Enhanced Yellow Fluorescent Protein (eYFP) and were transiently expressed in Arabidopsis protoplasts.”
Response: Thank you for pointing it out. We have revised the sentences with mistakes in the main text.
Reviewer 2 Report
The authors of this manuscript describe that, for the first time, they identified a transcription factor working not only in the nucleus but in the chloroplasts too.
They show that NAC102 localizes to the chloroplasts as well as the nucleus and specifically at the nucleoids. They present adequate data that NAC102 is guided by a N-terminal sequence to the chloroplasts and that NAC102 interacts with components of the two chroloplast RNA polymerases, using two different methods, yeast two hybrid and co-immuniprecipitation.
Finally they show that overexpression of NAC102 leads to a decrease in chroloplast gene expression, suggesting that NAC102 works as a transcriptional repressor.
The work is well planned and well presented.
Some minor points:
introduction would benefit if a little more information about NAC102 and/or the NAC family of transcription factors is provided
line 57 (Clough and Bent, 1998) should be a number and cited in the literature at the end (the citation is missing)
line 99 RNA isolation and qPCR
what tissue was used for the gene expression? is it the 3rd and 4th leaves like for chlorophyl measurement?
line 180 maybe add a reference to that sentence
in figures 1,2,3 could you provide some kind of quantitative data about the expression patterns seen in protoplasts?
in figure 4 could you state how many times the co-IP expreriment was repeated (biological repeats)
line 251 the authors state ''One possible reason is functional redundancy.''
the authors should provide more information about the NAC102 family members, their homology and information about their predicted chloroplast localization to support their hypothesis that its functional redundancy that masks the nac102 mutant phenotype
the authors should also provide in the supplementary data, the expression values of the two NAC102-eYFPNES lines
Author Response
Reviewer: 2
Comments and Suggestions for Authors
The authors of this manuscript describe that, for the first time, they identified a transcription factor working not only in the nucleus but in the chloroplasts too.
They show that NAC102 localizes to the chloroplasts as well as the nucleus and specifically at the nucleoids. They present adequate data that NAC102 is guided by a N-terminal sequence to the chloroplasts and that NAC102 interacts with components of the two chroloplast RNA polymerases, using two different methods, yeast two hybrid and co-immuniprecipitation.
Finally they show that overexpression of NAC102 leads to a decrease in chroloplast gene expression, suggesting that NAC102 works as a transcriptional repressor.
The work is well planned and well presented.
Response: We appreciate your positive comments on our manuscript and giving us constructive suggestions.
Some minor points:
introduction would benefit if a little more information about NAC102 and/or the NAC family of transcription factors is provided
Response: Thank you for your suggestion. We have introduced the biological function of NAC102 and cited the latest reference (Line 48-52).
line 57 (Clough and Bent, 1998) should be a number and cited in the literature at the end (the citation is missing)
Response: Thank you for pointing it out. We revised the format and added the citation [22].
line 99 RNA isolation and qPCR
what tissue was used for the gene expression? is it the 3rd and 4th leaves like for chlorophyl measurement?
Response: We are sorry for missing this information! We quantified the expression of chloroplast genes in both 8-d-old seedlings and the 3rd and 4th leaves of 3-week-old seedlings. The gene expression pattern in both tissues was similar. The presented data was obtained using the 8-d-old seedlings. We added the information in this section.
line 180 maybe add a reference to that sentence
Response: Thank you for your suggestion. We added the reference [31].
in figures 1,2,3 could you provide some kind of quantitative data about the expression patterns seen in protoplasts?
Response: Thank you for your suggestion. Our conclusion in this manuscript is that NAC102 localizes in both chloroplast and nucleus. We did not examine the dynamic change of their localization. Therefore, we believed that pictures were sufficient to support this conclusion.
in figure 4 could you state how many times the co-IP expreriment was repeated (biological repeats)
Response: Thank you for your comment. The CoIP assays were repeated twice with similar results.
line 251 the authors state ''One possible reason is functional redundancy.''
the authors should provide more information about the NAC102 family members, their homology and information about their predicted chloroplast localization to support their hypothesis that its functional redundancy that masks the nac102 mutant phenotype
Response: Thank you for your comment. Among the NAC family members, only NAC102 is predicted to localize in chloroplast. However, we found that many other transcription factors such as ARF2 (AT5G62000), YABBY1 (AT2G45190), WRKY3 (AT2G03340), bHLH014 (AT4G00870), and bZIP52 (AT1G06850) are predicted to localize in the chloroplast. We guess that other chloroplast-localized transcription factors may function similarly to NAC102 and mask the nac102mutant phenotype.
the authors should also provide in the supplementary data, the expression values of the two NAC102-eYFPNES lines
Response: Thank you for your comment. These data were shown in Figure S4B in the original version.